# Distribution and transmission of *M. tuberculosis* in a high-HIV prevalence city in Malawi: A genomic and spatial analysis

Melanie H. Chitwood[1]*, Elizabeth L. Corbett[2], Victor Ndhlovu[3], Benjamin Sobkowiak[1], Caroline Colijn[4], Jason R. Andrews[5], Rachael M. Burke[2,6], Patrick G.T. Cudahy[7], Peter J. Dodd[8], Jeffrey W. Imai-Eaton[9,10], David M. Engelthaler[11], Megan Folkerts[11], Helena R.A. Feasey[6,12], Yu Lan[1], Jen Lewis[8], Jennifer McNichol[4], Nicolas A. Menzies[13], Geoffrey Chipungu[14], Marriott Nliwasa[6,14], Daniel M. Weinberger[1], Joshua L. Warren[15], Joshua A. Salomon[16], Peter MacPherson[2,6,17,18‡], Ted Cohen[1‡*]

1 Department of Epidemiology of Microbial Diseases, Yale School of Public Health, New Haven, Connecticut, United States of America, 2 Clinical Research Department, Faculty of Infectious and Tropical Disease, London School of Hygiene and Tropical Medicine, London, United Kingdom, 3 School of Life Sciences and Health Professions, Kamuzu University of Health Sciences, Blantyre, Malawi, 4 Department of Mathematics, Simon Fraser University, Burnaby, British Columbia, Canada, 5 Division of Infectious Diseases and Geographic Medicine, Stanford University School of Medicine, Stanford, California, United States of America, 6 Public Health Group, Malawi-Liverpool-Wellcome Programme, Blantyre, Malawi, 7 Division of Infectious Diseases, Department of Internal Medicine, Yale School of Medicine, New Haven, Connecticut, United States of America, 8 Sheffield Centre for Health and Related Research, School of Medicine and Population Health, Sheffield, United Kingdom, 9 Center for Communicable Disease Dynamics, Department of Epidemiology, Harvard T.H. Chan School of Public Health, Boston, Massachusetts, United States of America, 10 MRC Centre for Global Infectious Disease Analysis, School of Public Health, Imperial College London, London, United Kingdom, 11 Translational Genomics Research Institute, Flagstaff, Arizona, United States of America, 12 School of Medicine, University of St Andrews, St. Andrews, United Kingdom, 13 Center for Health Decision Science and Department of Global Health and Population, Harvard T.H. Chan School of Public Health, Boston, Massachusetts, United States of America, 14 Helse Nord Tuberculosis Initiative, Department of Pathology, Kamuzu University of Health Sciences, Blantyre, Malawi, 15 Department of Biostatistics, Yale School of Public Health, New Haven, Connecticut, United States of America, 16 Department of Health Policy, Stanford University School of Medicine, Stanford, California, United States of America, 17 School of Health & Wellbeing, University of Glasgow, Glasgow, United Kingdom, 18 Bacterial and Respiratory Pathogens Department, Public Health Scotland, Glasgow, United Kingdom

‡ co-senior authors
* melanie.chitwood@yale.edu (MHC); theodore.cohen@yale.edu (TC)

## Abstract

Delays in identifying and treating individuals with infectious tuberculosis (TB) contribute to poor health outcomes and allow ongoing community transmission of *M. tuberculosis* (*Mtb*). Current recommendations for screening for tuberculosis specify community characteristics (e.g., areas with high local tuberculosis prevalence) that can be used to target screening within the general population. However, areas of higher tuberculosis burden are not necessarily areas with higher rates of transmission. We investigated the transmission of *Mtb* using high-resolution surveillance data in Blantyre, Malawi. We extracted and performed whole genome sequencing on mycobacterial DNA from cultured *M. tuberculosis* isolates obtained from culture-positive tuberculosis cases at the time of tuberculosis (TB) notification in Blantyre, Malawi between 2015-2019. We constructed putative transmission

**Data availability statement:** Whole genome sequencing data reported in this study will be available through NCBI under BioProject number PRJNA1221228.

**Funding:** MHC, ELC, VN, SB, CC, JRA, PGTC, PJD, DME, MF, YL, JL, NAM, JLW, JAS, TC acknowledge funding from the National Institute of Allergy and Infectious Diseases (NIAID)(R01AI147854)(https://www.niaid.nih.gov/). JWI-E acknowledges funding from the Bill & Melinda Gates Foundation (INV-006733, INV-002606)(https://www.gatesfoundation.org/) and the MRC Centre for Global Infectious Disease Analysis (reference MR/X020258/1), funded by the UK Medical Research Council (MRC)(https://www.ukri.org/councils/mrc/). This UK funded award is carried out in the frame of the Global Health EDCTP3 Joint Undertaking. The funders did not play any role in the study design, data collection and analysis, decision to publish, or the preparation of the manuscript.

**Competing interests:** JIE is an academic editor at PLOS Global Public Health. The authors have no other competing interests to declare.

networks identified using TransPhylo and investigated individual and pair-wise demographic, clinical, and spatial factors associated with person-to-person transmission. We found that 56% of individuals with sequenced isolates had a probable transmission link to at least one other individual in the study. We identified thirteen putative transmission networks that included five or more individuals. Five of these networks had a single spatial focus of transmission in the city, and each focus centered in a distinct neighborhood in the city. We also found that approximately two-thirds of inferred transmission links occurred between individuals residing in different geographic zones of the city. While the majority of detected tuberculosis transmission events in Blantyre occurred between people living in different zones, there was evidence of distinct geographical concentration for five transmission networks. These findings suggest that targeted interventions in areas with evidence of localized transmission may be an effective local tactic, but will likely need to be augmented by city-wide interventions to improve case finding to have sustained impact.

## Introduction

Tuberculosis (TB) is a major global health threat and a leading infectious cause of death. The World Health Organization's (WHO) End TB Strategy aims to reduce global tuberculosis incidence by 80% by 2030 from 2015 levels [1]. Rapid diagnosis and treatment, key pillars of the End TB Strategy, can reduce tuberculosis transmission by limiting the time an individual is infectious and potentially transmitting *M. tuberculosis* (*Mtb*). Passive case detection, which depends on individuals with tuberculosis seeking care, is insufficient to rapidly reduce *Mtb* transmission in most settings [2]. In high-burden settings, WHO recommends systematic screening for tuberculosis disease in communities [3]. However, there is inconsistent evidence to indicate whether screening decreases tuberculosis prevalence [4,5].

Targeting screening in areas where most transmission occurs may decrease TB prevalence [6,7], but identifying these areas using routinely collected data is challenging. Among newly infected individuals, the incubation period is variable and the risk of progressing to symptomatic disease is generally low [8]. Areas of high disease burden may therefore reflect higher risk of progression to active disease rather than higher risk of transmission [9]. This phenomenon may be more pronounced in settings with a high prevalence of human immunodeficiency virus (HIV) because people living with HIV are more likely to progress to active tuberculosis disease [10] and less likely to transmit *Mtb* to others [11]. Consequently, methods are needed that can identify areas of active transmission, which may not necessarily align with areas of high notification rates.

The increasing availability of whole genome sequencing (WGS) data, paired with methodological advances in transmission inference, has improved the ability to understand pathogen transmission dynamics [8,12,13], information that is critical for the design of targeted active case finding efforts. Several studies have leveraged these types of data to characterize spatial patterns in tuberculosis burden and *Mtb* transmission [14–19], but few have been conducted in cities with high rates of TB/HIV co-infection [20].

The aim of this study was to describe patterns of TB transmission in an endemic setting using high-resolution surveillance and WGS data. In this study, we collected and sequenced mycobacterial specimens from individuals diagnosed with culture-positive tuberculosis in Blantyre, Malawi between 2015 and 2019. We used WGS data to infer networks of transmission; we paired those findings with geographical coordinates (GPS) of patient home locations to identify local transmission of specific strains and describe patterns of transmission between

administrative areas. We hypothesized that whole genome sequencing would provide novel insights into *Mtb* transmission dynamics in a city with a high prevalence of both tuberculosis and HIV.

## Methods

### Study setting and population

Blantyre is a city in southern Malawi, with a population of approximately 800,000 [21]. It is the second largest city in Malawi and is the nation's industrial and commercial capital. Over half of the population live in areas without access to basic municipal services and 43% of city land is considered unplanned or rural [22]. Blantyre is a hilly city, and its varied topography creates distinct neighborhoods separated by ridges and valleys.

This retrospective study included people identified through passive case detection and diagnosed with tuberculosis in Blantyre, Malawi between 1 January 2015 and 31 December 2019. All people with notified tuberculosis in Blantyre were registered in the ePAL (electronic Participant Locator) system [23–25]. ePAL is an app-based data entry platform for the collection of patient information combined with an electronic case report form with high resolution satellite maps and community-identified points of interest [25]. Data available in ePAL include age, sex, diagnosing clinic, microbiology results (acid-fast bacillus [(AFB]) smear and Xpert MTB/RIF), symptom history and duration, tuberculosis classification (pulmonary, extra-pulmonary), HIV and antiretroviral therapy (ART) status, HIV clinic (if applicable), presence of known TB exposures (e.g., household contacts), locations of the three most recent clinics attended, number of hospital admissions within the year preceding diagnosis, and a range of poverty indicators. The patient's current home location, selected via touch-screen and converted in-app to GPS coordinates [25], is also available. The authors of this study had access to data that could identify individual participants, including age, gender, and home GPS coordinate data. The authors were given access to all data in January, 2021.

We estimated tuberculosis notification rates at a resolution of 500 m² grid cells. Population denominators were calculated from WorldPop [21] 2020 data and aggregated from the original resolution of 100 m². We also present grid-specific HIV prevalence estimates, which were derived from two national HIV prevalence studies, one Blantyre-specific prevalence study, and antenatal prevalence data. These data were combined in a Bayesian model to derive highly spatially resolved HIV prevalence estimates, as described previously [26].

### Ethics statement

The study protocols were reviewed and approved by the University of Malawi College of Medicine Research and Ethics Committee (#P.12/18/2556), the London School of Hygiene and Tropical Medicine (#16228-4), and Yale (#2000028431). Oral consent was provided by people registering for TB treatment for electronic data capture, including recording of household co-ordinates. Oral consent and assent were used for the latter since the electronic register data capture was conducted as part of normal clinical practice by District TB Officers. Jr

### Laboratory regrowth, DNA isolation, and whole genome sequencing

We set out to re-culture and sequence samples from all culture-positive cases notified over the five-year study period. Study isolates were thawed from -80°C and cultured in liquid Middlebrook 7H9 media using the BD BACTECT™ MGIT™ 960 system, then subcultured on Löwenstein–Jensen medium at 37°C to obtain pure colonies. A minimum of 1 μg *Mtb DNA* (either 100μl of 10000 ng/ml, or 40μl of 25000 ng/ml, etc) was manually extracted following standard

operating procedures as previously described [27] and validated in-country, with DNA stored at -20°C before shipping for sequencing at TGen in Flagstaff, Arizona.

Sequencing libraries were constructed using either Illumina's DNA Prep kit or Watchmaker's DNA Library Prep Kit with Fragmentation. Whole genome sequencing (WGS) was performed on Illumina NextSeq550 or NextSeq1000 to produce paired-end 150 bp reads. A phiX (Illumina) sequencing control was spiked into each run at 1% of the total library to be sequenced to facilitate run performance monitoring. Raw sequencing FASTQ files were checked for non-Mycobacterium DNA using Kraken; [28] isolates containing > 80% *Mtb* reads were retained and non-*Mtb* reads filtered out using a custom script. Sequence reads were mapped to the H37Rv reference strain (GenBank accession number NC_000962.3) with BWA v.0.7.17 'mem' [29], removing sequences with < 80% mapping to the reference strain and <50x average coverage.

Variant calling was carried out using GATK v.4.4.0.0 'HaplotypeCaller' and 'GenotypeGVCFs' [30]. Single nucleotide polymorphisms (SNPs) were filtered to remove sites with low quality (Q < 20), low read depth (DP < 5), or high proportion missingness (missing call in ≥ 10% of isolates). Sites showing more than one allele (mixed sites) were assigned the majority allele where ≥ 90% of reads agreed, otherwise these were assigned an ambiguous character 'N'. Finally, sequences with a high likelihood of mixed infection identified using MixInfect [31] were removed. *In silico* lineage prediction and drug resistance profiling was carried out on the remaining isolates using TB-Profiler v.5.0.1 [32].

## Phylogenetic reconstruction

A multi-sequence alignment of concatenated SNPs was used to produce phylogenetic trees. SNPs in repetitive regions and known microbial resistance-associated and PE/PPE genes were removed to account for potential homoplasy that may confound phylogenetic reconstruction (S1 Table). IQ-tree v.2.2.2.6 [33] was used to construct a maximum-likelihood phylogeny of all isolates, with the '-m TEST' parameter used to determine the optimal nucleotide substitution model of Kimura's model with unequal base frequencies (K3Pu+F).

## Transmission inference

Transmission networks and the probability of person-to-person transmission among sequenced cases were inferred using a two-step process. First, we identified preliminary, broad clusters of sequences using a 50 SNP threshold and constructed timed phylogenies with BEAST2 v.2.7.5 [34]. We used a relaxed lognormal substitution rate and constant coalescent population tree prior, along with adapting the XML file to include a correction for invariant sites. We ran the model for $2 \times 10^8$ Markov chain Monte Carlo (MCMC) iterations or until convergence was achieved and an adequate number of posterior samples were collected, demonstrated by the collected samples from all parameters reaching an effective sample size (ESS) of greater than 200 after a 20% burn-in was discarded.

Second, we ran TransPhylo [35] on each broad cluster to identify transmission networks. We used the implementation of TransPhylo with simultaneous inference of multiple trees [14] to account for phylogenetic uncertainty by taking a random sample of 50 posterior trees from the BEAST2 output, discarding the first 20% as burn-in. We assumed a prior gamma generation time distribution (α =1.3, β = 0.3) and a prior gamma sampling time distribution (α =1.1, β = 0.4), as has been previously applied for *Mtb* transmission reconstruction [35–37]. We ran the model for $1 \times 10^5$ MCMC iterations using a fixed within-host coalescent parameter of 100/365 and a beta sampling proportion distribution (α = 2, β = 20) that updated through the runs. This produced a predicted pairwise probability of direct transmission between isolates in

clusters, and all isolates that were not present in the same broad cluster were *de facto* assigned a pairwise transmission probability of 0.

Transmission networks predicted by TransPhylo for each broad cluster with the highest probability were further refined. Where inferred networks were linked by more than three non-sampled hosts, we considered these as separate networks. This allowed us to identify putative transmission networks while accounting for missing cases.

### Identification of spatial foci of transmission

For each putative transmission network with at least five cases, we investigated spatial areas where individuals with tuberculosis had relatively higher likelihoods of transmission to or from others in the area, based on individuals' reported home locations. We called these areas spatial foci of transmission, and we identified them using a non-parametric distance-based mapping (DBM) [38] approach implemented in the R package hotspotr [39]. Using hotspotr, we divided the city into a 100 by 100 grid of cells (each cell is 194 m by 144 m). The analysis then proceeded for each transmission network. First, we selected a transmission network to analyze. Second, we used the home location of sequenced TB cases that were not part of that transmission network to calculate the expected number of cases within each grid cell. Third, we calculated the risk that there are more cases belonging to the transmission network in a given grid cell than expected, assigning a score between 0 (no spatial aggregation of individuals from the same transmission network) and 1 (highest risk of spatial aggregation). We repeated these steps for each transmission network with five or more individuals. Any grid cell with a score $\geq 0.95$ was considered a spatial focus of transmission, and we used a narrow window (width = 0.01) to smooth results across grid cells [39]. Note that this method allows that there may be multiple foci of transmission for a single transmission network.

### Estimation of factors associated with transmission

We used a dyadic regression model to understand the factors associated with pairs of individuals (1) belonging to the same transmission network and (2) having more closely related *Mtb* isolates (conditional on belonging to the same network). To answer the first question, we fitted a single logistic regression model using a binary outcome of whether two individuals were in the same network. We included several predictors (age, sex, HIV status, clinic where HIV care is received (if applicable), clinic where TB diagnosis was made), and we included the distance between homes as two effects in the model; continuous distance between cases that were close together (< 6 km) and cases that were further apart ($\geq 6$ km). We chose to model distance in this way because we believe the association between distance and cluster membership may be stronger at shorter distances; we performed model comparisons using AIC, and found that two effects outperformed a single linear effect, and a 6 km threshold outperformed other choices of threshold.

To answer the second question (i.e., which factors are associated with two individuals in the same network having more similar strain sequences), we performed a separate analysis in which we fitted a negative binomial regression model with a logit link for each transmission network with at least 10 cases (i.e., $\binom{10}{2} = 45\text{pairs}$ ), and we modeled the number of SNPs by which pairs differed. This allowed us to investigate predictors of genetic similarity of isolates. We used the same set of covariates as above, and also included the zone in which the individuals resided. We excluded predictors if there was missingness among any individuals within the network or if there were fewer than four pairs of individuals in any level of the predictor.

We implemented these models using the R package GenePair [40]. Regression models with dyadic outcomes will produce effect estimates with erroneously reduced uncertainty if the correlation between dyadic outcomes is not considered [40,41]. GenePair addresses this potential problem by including spatially structured individual-level random effect parameters that induce correlation between the dependent variables [40]. For all analyses, we based inference on samples from the joint posterior distribution, removing 10,000 iterations of burn-in and thinning the remaining 25,000 by a factor of 5 to reduce correlation in the posterior samples.

### Transmission flow analysis

Using the posterior transmission probabilities obtained from TransPhylo, we computed the number of transmission events between and within each of the seven zones (we mapped a sequenced TB case to a zone based on their reported home location), assuming one possible infector per infected individual. That is, the individual with the highest posterior probability of transmission is taken to be the infector of each recipient. We then assign infector-recipient pairs to their respective zones. The transmission flows from zone a to zone b, adjusted for sampling are given by

$$\pi_{ab} = \frac{\left( \dfrac{n_{ab}}{\xi_a \xi_b} \right)}{\left( \sum_{c,d} \dfrac{n_{cd}}{\xi_c \xi_d} \right)}$$

where $n_{ab}$ denotes the number of transmission events from zone a to zone b and $\xi_a$ denotes the sampling fraction in region a (S2 Table). We compared this approach to a previously published method to estimate flows based on transmission trees [42]; our results were consistent with this established approach (S2 Fig).

## Results

Between January 2015 and December 2019, 12,238 tuberculosis disease episodes were notified and recorded in ePAL. The mean tuberculosis notification rate over the study period was 279 per 100,000 population per year, which declined from 299 per 100,000 in 2015 to 225 per 100,000 in 2019. The notification rate varied across city zones, and grid cells with high notification rates abutted areas of apparent lower burden (Fig 1A). Among individuals notified in ePAL, most were new cases (10,608; 87%), most were male (7,438; 61%), the median age was 35 years (IQR: 26, 43), and most were living with HIV (7,860; 64%). Areas of higher HIV prevalence in the general populations in Zone 2 and Zone 6 overlapped with areas with high tuberculosis notification rates (Fig 1B). Population coverage of ART is estimated to be greater than 50% [43].

### Whole genome sequencing analysis

Over the study period, 8,386 (68.5% of all TB cases) individuals received a TB culture test and 3,856 (31.5% of all TB cases) individuals tested positive by tuberculosis culture. Among culture-positive specimens, 1,333 (34.6%) were available for whole genome sequencing, and 1,009 (26.2%) samples could be matched to patient clinical data (S3 Fig). Among sequenced isolates, 861 (64.6% of available isolates, 22.3% of positive cultures) passed quality control checks and 717 (53.8% of available isolates, 18.6% of positive cultures) were found to be pure (non-mixed) samples, which were included in the final sample dataset. Home location data were available for 707 (99%) of these cases (Fig 1C). The population characteristics of cases

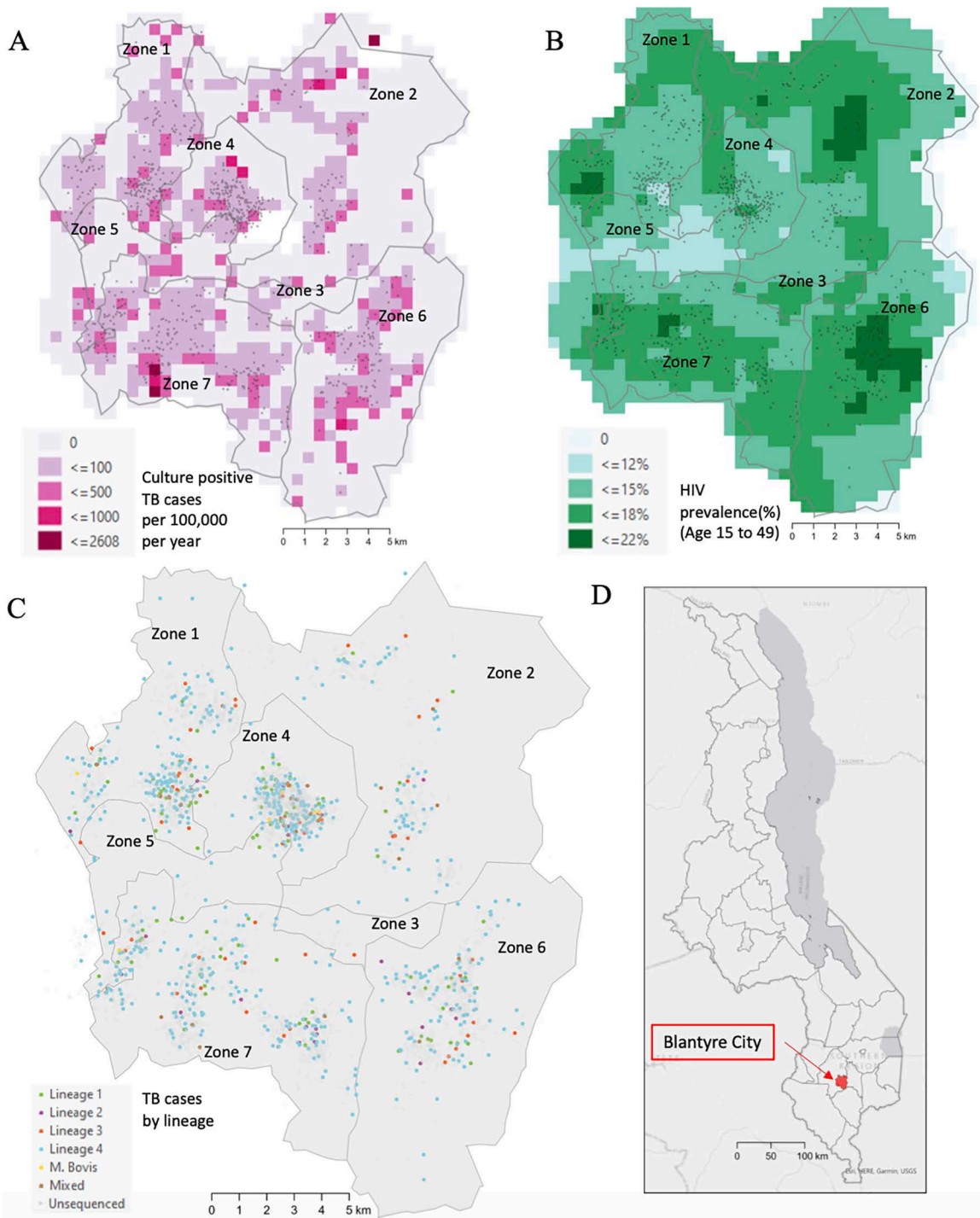

**Fig 1. Map of Blantyre, Malawi.** (A) Culture positive tuberculosis case notifications per 100,000 population per year over the study period; (B) HIV prevalence estimates for the city population (with and without TB-coinfection), aged 15-49 years; (C) all notified tuberculosis cases, colored by major lineage or shown in grey if no sequencing data were available; (D) inset map indicating the location of Blantyre in Malawi. All points are based on reported home locations. Points have been jittered for privacy. Zones drawn by authors based on existing administrative boundaries. Base map citation: Humanitarian Data Exchange, Accessed May 1 2023, https://data.humdata.org/dataset/cod-ab-mwi CC-BY-IGO.

included in the final sample dataset can be found in Table 1. We compare all TB cases, culture-positive TB cases, and cases included in the final sample dataset in S3 Table.

A freezer failure, during which alarms were not acted on (due to COVID-19 lockdowns) occurred in 2020, resulting in lower-than-expected sequencing yield for affected frozen *Mtb* isolates. The fraction of individuals with a successfully sequenced diagnostic specimen did not differ meaningfully by sex, age, HIV status (Table 1), or city zone of residence (S4A Table). However, the fraction of successfully sequenced culture positive cases varied by year, ranging from 12% to 33% (S4B Table).

Most isolates included in the final sample dataset belonged to lineage 4 (Euro-American, 72%) followed by lineage 1 (Indo-Oceanic, 14%) (Fig 2), similar to previously reported studies in Blantyre [44]. The diversity of strains differed markedly by lineage; lineages 1 and 4 had larger pairwise SNP distances, on average, as compared to lineages 2 and 3 (S4 Fig). The proportion of sequences collected from each lineage remained consistent throughout the study period (S5 Fig). There was a low prevalence of drug resistance, with 94% (676/717) of samples susceptible to all antimicrobials. We identified 2 rifampin mono-resistant, 22 isoniazid mono-resistant strains, and 4 multidrug resistant strains (< 1%).

## Transmission inference

There were 393 isolates (56%) that belonged to one of 130 transmission networks inferred with TransPhylo and had an associated home location. The average pairwise SNP distance within a network was 4.5 (IQR: 1, 7). Most transmission networks contained only one pair of individuals (87/130; 67%); 13 transmission networks comprised five or more individuals (10%)(S6 Fig) and three transmission networks comprised 10 or more individuals (2%) (Fig 3). The largest transmission network contained 25 individuals. There were 50 pairs of individuals with a high probability of direct transmission (probability ≥ 0.5) and a further 97 pairs with a moderate probability of direct transmission (probability ≥ 0.25).

We also attempted to identify household transmission events. In total, 270 individuals (2.2% of the TB notifications over the study period) reported a household contact with TB. The final sample dataset included 44 of these individuals (6.1% of the dataset); among these

Table 1. Characteristics of cases with positive culture results.

| Characteristic | Culture Positive, Not Sequenced N = 3,148 | Final Sample dataset N = 717 |
|---|---|---|
| Sex | | |
| Male | 2,046 (65%) | 496 (69%) |
| Female | 1,102 (35%) | 221 (31%) |
| Age | | |
| Median | 34 | 33 |
| IQR | (27,40) | (26,39) |
| HIV status | | |
| Positive, on ART | 1,663 (53%) | 380 (53%) |
| Positive, Not on ART | 194 (6.2%) | 50 (7.0%) |
| Negative | 1,220 (39%) | 269 (38%) |
| Not Collected | 71 (2.3%) | 18 (2.5%) |
| Patient Type | | |
| New | 2,754 (87%) | 647 (90%) |
| Relapse | 312 (9.9%) | 61 (8.5%) |
| Other | 82 (2.6%) | 9 (1.3%) |

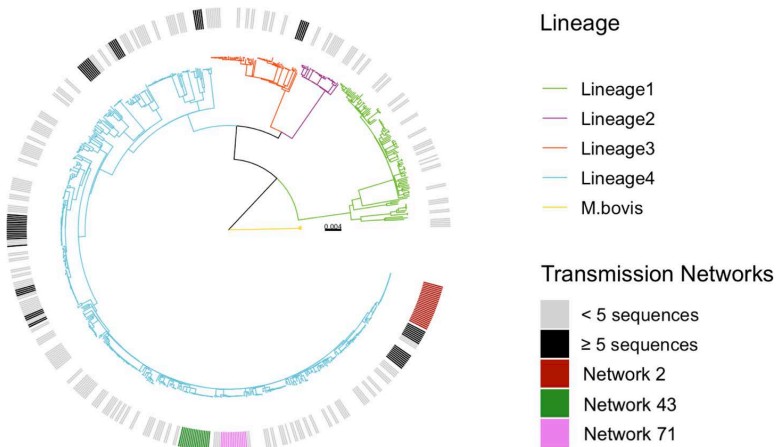

**Fig 2. A maximum likelihood phylogeny illustrating the genetic relatedness of the 717 isolates included in the study.** Branch lengths are colored by major *Mycobacterium tuberculosis* complex lineage and scaled by substitutions per site. Outer band denotes transmission network membership; the three largest networks are shown in maroon (Network 2), green (Network 43), and pink (Network 71).

individuals, 21 (47.7%) belonged to any transmission network. Individuals who reported having a household contact with TB were not more likely to belong to a transmission network than individuals who did not report a household contact (two-sample proportion test, p = 0.36). In addition, while 19 individuals could plausibly belong to the same household as another individual in the final sample dataset (i.e., home GPS coordinates were within 50 meters), none of these plausible household pairs belonged to the same transmission network.

## Identification of spatial foci of transmission

In distance-based mapping (DBM) of 13 transmission networks with ≥ 5 individuals (S5 Table), we detected spatial foci of recent transmission in five transmission networks (Fig 4). For the largest transmission network (network #2, n = 25), we did not detect any spatial aggregation of cases, suggesting that it was widespread throughout the city.

Transmission network #47 (n = 17) was the largest network for which we identified a spatial focus. Most individuals in this network were people living with HIV (n = 10, 67%) and the spatial focus of transmission was in a high HIV prevalence area. Network #71 was the second largest transmission network, and had a spatial focus on the periphery of the city where there was a high case notification rate. Only 20% (n = 3) of individuals within network #71 were people living with HIV.

Not all members of a transmission network with a spatial focus of transmission lived in or near the focus we identified (S5 Table). In network #80, 75% of network members (n = 6) lived inside the focus, and, in network #1, 60% of network members (n = 3) lived within 1km of the focus boundary (Fig 3A). Overall, of the 393 individuals with TB belonging to a transmission network, 252 (64%) lived within 1 km of a transmission focus.

## Estimation of factors associated with transmission

We performed a logistic regression to estimate factors associated with belonging to the same transmission network. We analyzed individuals who had both WGS data and home location data (n = 707) and found that two individuals had a higher odds of belonging to the same putative transmission network if they shared a tuberculosis diagnostic clinic (adjust odds

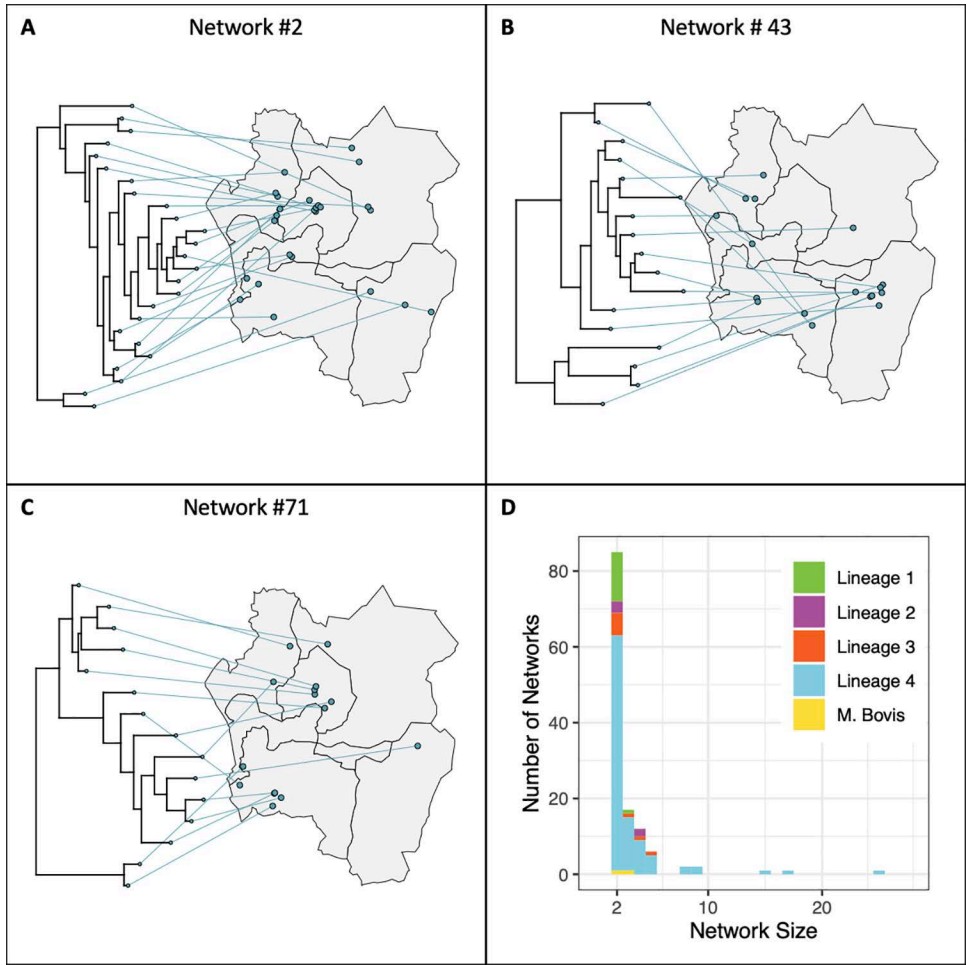

**Fig 3. Putative transmission networks.** (A–C) Time-resolved phylogenies with taxa linked to case home location. Each panel contains a different transmission network. The three networks with 10 or more individuals are shown; all three networks have strains belonging to lineage 4. Points have been jittered for privacy. (D) Distribution of transmission network sizes (number of sampled hosts within the network), colored by main lineage. Zones drawn by authors based on existing administrative boundaries. Base map citation: Humanitarian Data Exchange, Accessed May 1 2023, https://data.humdata.org/dataset/cod-ab-mwi CC-BY-IGO.

ratio (aOR) 1.58 95% highest posterior density interval (HPD) 1.13, 2.09) (Fig 5A). A 1 km increase in the distance between home location was associated with a 0.82 (0.77, 0.87) aOR of belonging to the same cluster for distances up to 6 km. Each 1 km increase in distance above the 6 km threshold was associated with a 0.91 (0.88, 0.93) aOR. We did not find a significant association between shared HIV clinic exposure (aOR 0.87, 95% HPD 0.57, 1.23), or HIV status (aOR for HIV positive pairs: 0.95 [0.65, 1.30]; aOR for HIV negative pairs: 1.34 [0.92, 1.86]) and belonging to the same putative transmission network. Because there was very high coverage of antiretroviral therapy among HIV positive individuals in this study (91%), we did not include ART as a covariate in our model.

We also performed a negative binomial regression to understand factors associated with closer genetic relatedness of isolate strains. We analyzed the three largest transmission networks (#2, #43, and #71) in separate analyses (Fig 5B). Individuals in transmission network #2 whose TB was diagnosed at the same clinic had sequences with smaller SNP differences on

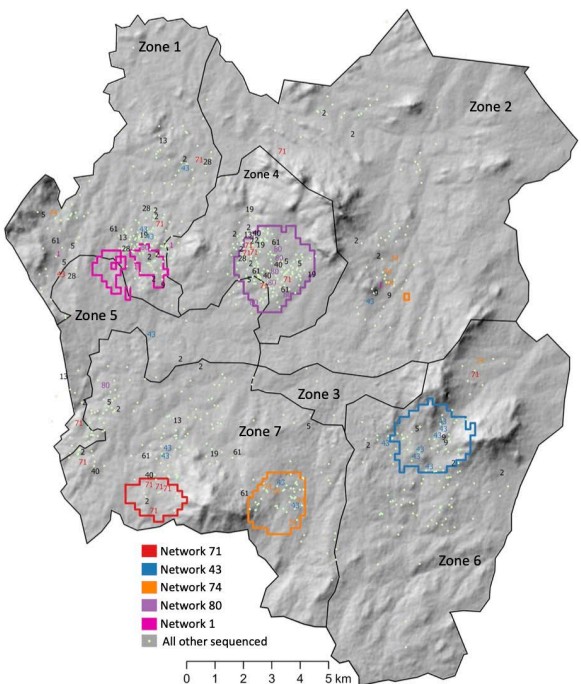

**Fig 4. Foci of recent transmission.** Foci identified using a distance based mapping (DBM) approach based on home address, applied to transmission networks with 5 or more individuals. Circled regions represent areas where the risk of spatial aggregation for a network is greater than 95% (spatial foci). A single transmission network may have multiple spatial foci. Individuals belonging to a transmission network with a spatial focus may reside outside the focus. Points have been jittered for privacy. Zones drawn by authors based on existing administrative boundaries. Base map citation: Humanitarian Data Exchange, Accessed May 1 2023, https://data.humdata.org/dataset/cod-ab-mwi CC-BY-IGO. Topographic information: Republished from Africa GeoPortal ("Malawi SRTM DEM 30meters") under a CC BY license, with permission from Patrick Kabatha, original copyright 2017. Accessed April 24, 2024. https://www.africageoportal.com/datasets/rcmrd::malawi-srtm-dem-30meters/about.

average (aRR 0.58, 95% HPD 0.28, 0.93). We did not identify statistically significant effects of any covariates on SNP differences in networks #43 and #71.

## Transmission flows analysis

We estimated between and within zone rates of transmission based on transmission pairs identified using TransPhylo (Fig 6). We found that 70% of transmission within the city occurred between, rather than within, zones. Zone 7 in the south-west of the city had the highest percentage of within-zone transmission, followed by Zone 4 in the center of the city. The highest rates of between zone transmission were between Zone 1 and Zone 7 and between Zone 4 and Zone 7. We estimate that 68% of all tuberculosis transmission in Blantyre could be attributed to infectious individuals from Zone 1, Zone 4, or Zone 7. Finally, there was effectively no transmission flow through Zone 3, a mostly industrial zone in the center of the city with a smaller residential population relative to other zones.

## Discussion

This analysis presents novel insights into transmission patterns of *M. tuberculosis* in Blantyre, Malawi between 2015 and 2019, a period when treatment coverage for HIV was being rapidly scaled up and TB notifications were falling. In phylogenetic analyses, 56% of sequenced isolates could be mapped to a putative transmission network. We identified two main

A

Odds that a pair of individuals belong to the same cluster

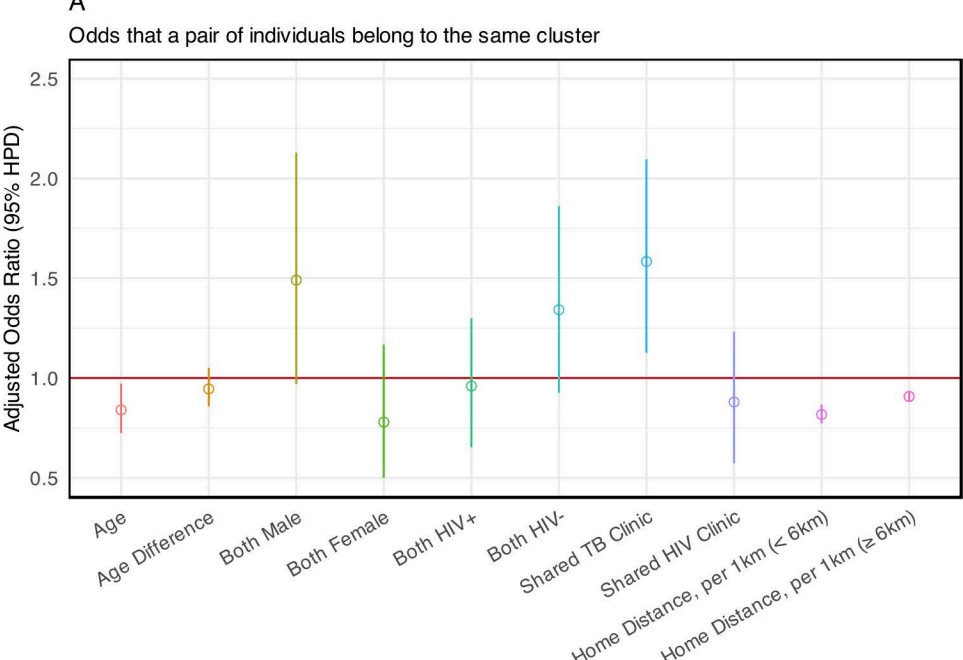

B

Risk that a pair of cases have isolates with fewer SNP differences

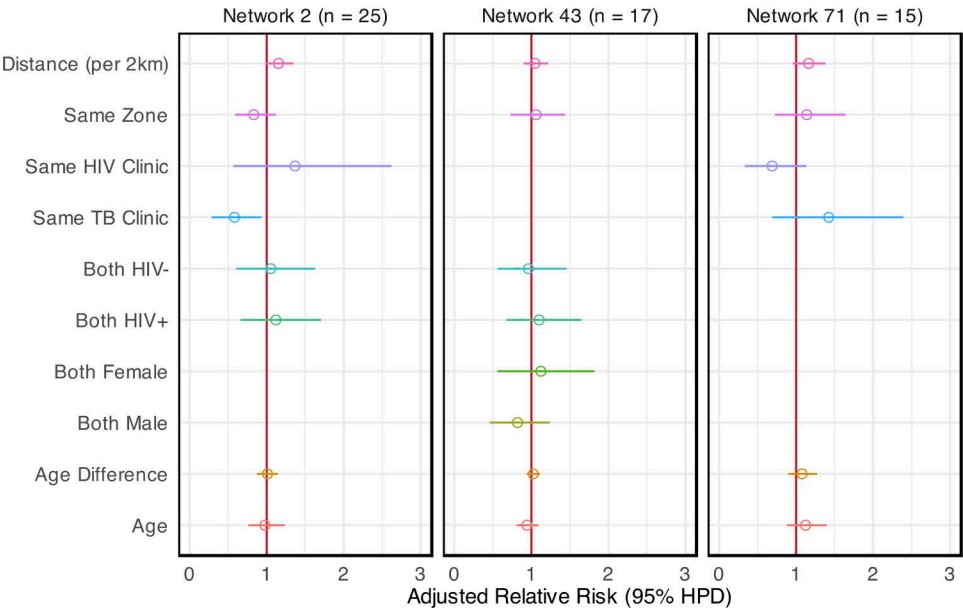

**Fig 5. Factors associated with transmission.** (A) Effect of covariates on the odds that case pairs belong to the same putative transmission cluster. Adjusted Odds Ratio > 1 is associated with increased odds of belonging to the same putative transmission cluster. (B) Effect estimates of covariates on SNP difference in isolates from case pairs within the same transmission network. Adjusted Rate Ratio < 1 is associated with smaller SNP differences on average; smaller SNP differences are associated with increased likelihood of direct transmission between case pairs. Gender is excluded in the analyses of network 2 and network 71 because only three individuals were female. HIV status is excluded in the analysis of network 71 because only three individuals were people living with HIV.

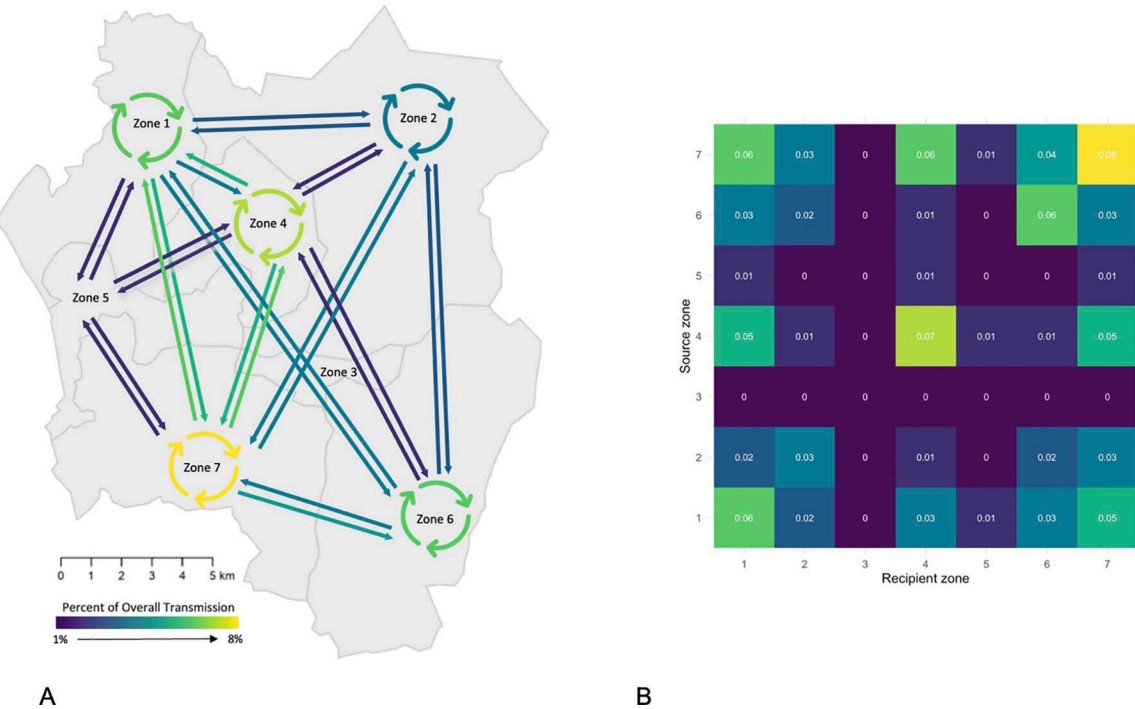

**Fig 6. Estimates of the percent of overall transmission occurring within- and between-zones of Blantyre.** (A) Map of estimates; arrows point from source zone to recipient zone, and cyclic arrows represent within-zone transmission. Arrows are omitted if there was no evidence of within-zone transmission (e.g., Zone 3) or if there was no evidence of between-zone transmission (e.g., Zone 5 to Zone 3). (B) Grid of estimates, with source region on the y-axis and receiving region on x-axis. Zones drawn by authors based on existing administrative boundaries. Base map citation: Humanitarian Data Exchange, Accessed May 1 2023, https://data.humdata.org/dataset/cod-ab-mwi CC-BY-IGO.

characteristics of transmission with implications for the design of public health interventions. First, despite evidence that Blantyre's tuberculosis epidemic is receding, most detected transmission events occurred across city zones. Second, we identified a sizeable minority of transmission events that occurred in local outbreaks in distinct areas of Blantyre. Geographical location of residence has long been known to be linked to tuberculosis epidemiology, with poor-quality housing, air pollution, undernutrition, crowding, and suboptimal access to healthcare as key drivers of incidence. While the residential proximity of two individuals is an important predictor of membership in the same transmission network (Fig 5A), we found that the majority of transmission events occurred between individuals residing in different zones of the city (Fig 6). High rates of inter-zone transmission are consistent with findings in other high TB/HIV burden settings [18], and indicate that high levels of population mixing may play a role in sustaining Blantyre's tuberculosis epidemic [45].

In this study, five of the thirteen large transmission networks were associated with a spatial focus of transmission, and none of these foci overlapped or abutted each other (Fig 4). This finding is distinct from patterns reported from other genomic analyses of tuberculosis in urban settings [20,39], and raises new questions about the projected impact of targeted interventions in locations with complex multifocal epidemics. We note that a limitation of this analysis is that spatial foci were identified based only on the home location of individuals with TB; we did not have data about social activity locations where transmission may occur, such as schools, workplaces, or other congregate settings. However, we found that receiving a TB diagnosis at the same clinic was associated with a greater odds of belonging to the same

transmission network, independent of home location of individuals. While this could be explained by transmission within healthcare settings, it could also reflect a shared neighborhood effect that was not completely captured by home location since people living in the same neighborhood would be more likely to attend the same clinic.

An important limitation of this study is the relatively low fraction of sequencing (18% of culture-positive isolates over the study period). Diagnostic isolates were available for approximately one-third of eligible cases, and lab contamination further reduced the number of sequenced isolates. This poses a challenge for inferring transmission networks, as most existing methods have been designed for more densely sampled outbreaks, and it is likely that sequenced isolates that were not within transmission networks in our analysis may actually have been linked with other isolates that were just not available or successfully sequenced. We address this limitation using two approaches: first, we accounted for this by allowing for a higher number of unsampled intermediate cases in transmission networks which introduced additional uncertainty to our transmission inference as expected. Second, we focused our analysis only on the largest transmission clusters which reduces the sensitivity of our analysis to the expected misclassification of isolates in small (i.e., 2-3 cases) transmission networks as unique isolates. We also note that 144 sequences were excluded due to evidence of polyclonal infections, which reduced the fraction of cases which could be included in this study because of the challenge in including these types of complex samples in phylogenetic reconstruction and transmission analysis.

Whole genome sequencing data provide information about local transmission and help to identify geographic areas where transmission is highest. Focused interventions to interrupt transmission could be effective at reducing incidence in these areas, and previous models have suggested that interventions targeted to a single transmission focus may have broader benefits to reducing incidence in an entire city [6]. However, given the interconnectedness of transmission in Blantyre, the degree to which targeting interventions to small numbers of geographic foci will have these positive 'spill-over' effects in unclear. Transmission models calibrated to the spatial transmission patterns estimated in this study provide an attractive next step for estimating the potential health benefits of targeting interventions to transmission foci. In the absence of such estimates, our results suggest that TB control efforts should focus on improving access to tuberculosis services through primary clinics citywide, and that supplementing these general improvements with targeted active case finding should be considered as a secondary tactic.

## Supporting information

**S1 Fig. Zones of Blantyre with ward boundaries.** Zone 7 has been expanded to include a small area outside the city boundary with a high TB notification rate (unlabeled ward between Green Corner Ward and Soche West Ward). Zones drawn by authors based on existing administrative boundaries. Base map citation: Humanitarian Data Exchange, Accessed May 1 2023, https://data.humdata.org/dataset/cod-ab-mwi CC-BY-IGO.
(TIFF)

**S2 Fig. Comparison of transmission event summation approach to existing ('Flows' on the y-axis) to a previously described transmission flow analysis ('PhyloFlows' on the x-axis).**
(TIFF)

**S3 Fig. (A) Fraction of cases that were culture-positive and sequenced over time and (B) fraction of sequenced cases belonging to a transmission network ("clustered") over time.**
(TIFF)

**S4 Fig. Pairwise SNP distances, by lineage.**
(TIFF)

**S5 Fig. Proportion of sequences belonging to the four major MTBC lineages and *M. bovis*.**
(TIFF)

**S6 Fig. Time-resolved transmission networks with 5-10 sampled cases.**
(TIFF)

**S1 Table. Sites that were excluded from the multi-sequence alignment.**
(XLSX)

**S2 Table. Population size estimates and sequencing coverage by zone.**
(XLSX)

**S3 Table. Comparison of TB cases, culture-positive TB cases, and cases included in the final sample dataset.**
(XLSX)

**S4 Table. Characteristics of sequenced and not sequenced culture positive TB cases.**
(XLSX)

**S5 Table. Summary of transmission networks used in the DBM analysis.**
(XLSX)

**S1 Data. Deidentified individual-level data used in this analysis (excluding home location and clinic information).**
(CSV)

**S1 Code. Example BEAST2 code to estimate time-calibrated phylogenetic trees.**
(XML)

## Author contributions

**Conceptualization:** Elizabeth L Corbett, Joshua A Salomon, Peter MacPherson, Ted Cohen.

**Data curation:** Elizabeth L Corbett, Victor Ndhlovu, Patrick G.T. Cudahy, David M Engelthaler, Megan Folkerts, Geoffrey Chipungu, Marriott Nliwasa, Peter MacPherson.

**Formal analysis:** Melanie H. Chitwood, Benjamin Sobkowiak, Yu Lan, Jennifer McNichol, Joshua L Warren.

**Funding acquisition:** Elizabeth L Corbett, Joshua A Salomon, Peter MacPherson, Ted Cohen.

**Investigation:** Melanie H. Chitwood, Elizabeth L Corbett, Benjamin Sobkowiak, Caroline Colijn, Jason R Andrews, Rachael M Burke, Peter J Dodd, Jeffrey W Imai-Eaton, Helena R.A. Feasey, Jen Lewis, Nicolas A Menzies, Daniel M Weinberger, Joshua L Warren, Joshua A Salomon, Peter MacPherson, Ted Cohen.

**Methodology:** Caroline Colijn, Jennifer McNichol, Joshua L Warren.

**Project administration:** Elizabeth L Corbett, Joshua A Salomon, Peter MacPherson, Ted Cohen.

**Resources:** Elizabeth L Corbett, Ted Cohen.

**Software:** Caroline Colijn, Joshua L Warren.

**Supervision:** Elizabeth L Corbett, Peter MacPherson, Ted Cohen.

**Validation:** Melanie H. Chitwood, Elizabeth L Corbett, Peter MacPherson, Ted Cohen.

**Visualization:** Melanie H. Chitwood, Benjamin Sobkowiak, Yu Lan.

**Writing – original draft:** Melanie H. Chitwood, Benjamin Sobkowiak, Yu Lan, Peter MacPherson, Ted Cohen.

**Writing – review & editing:** Melanie H. Chitwood, Elizabeth L Corbett, Victor Ndhlovu, Benjamin Sobkowiak, Caroline Colijn, Jason R Andrews, Rachael M Burke, Patrick G.T. Cudahy, Peter J Dodd, Jeffrey W Imai-Eaton, David M Engelthaler, Megan Folkerts, Helena R.A. Feasey, Yu Lan, Jen Lewis, Jennifer McNichol, Nicolas A Menzies, Geoffrey Chipungu, Marriott Nliwasa, Daniel M Weinberger, Joshua L Warren, Joshua A Salomon, Peter MacPherson, Ted Cohen.

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
