## [Decision Letter · Decision Letter 0]

30 Sep 2024

PGPH-D-24-01419Distribution and transmission of M. tuberculosis in a high-HIV prevalence city in Malawi: a genomic and spatial analysisPLOS Global Public Health

Dear Dr. Chitwood,

Thank you for submitting your manuscript to PLOS Global Public Health. After careful consideration, we have decided that your manuscript does not meet our criteria for publication and must therefore be rejected.

We are sorry that we cannot be more positive on this occasion. We very much appreciate your wish to present your work in one of PLOS's Open Access publications. Thank you for your support, and we hope that you will consider PLOS Global Public Health for other submissions in the future.

Yours sincerely,

Gagandeep Singh, M.D.

Academic Editor

---

## [Decision Letter · Decision Letter 1]

22 Dec 2024

PGPH-D-24-01419R1

Distribution and transmission of M. tuberculosis in a high-HIV prevalence city in Malawi: a genomic and spatial analysis

Dear Dr. Chitwood,

Thank you for submitting your manuscript to PLOS Global Public Health. After careful consideration, we feel that it has merit but does not fully meet PLOS Global Public Health’s publication criteria as it currently stands. Therefore, we invite you to submit a revised version of the manuscript that addresses the points raised during the review process (see below for reviews).

We look forward to receiving your revised manuscript.

Kind regards,

Madhukar Pai, MD, PhD

Editor-In-Chief

Journal Requirements:

Additional Editor Comments (if provided):

I would like to see a more detailed discussion of the study limitations, given the feedback by the reviewers.

Reviewers' comments:

Reviewer's Responses to Questions

**Comments to the Author**

Does this manuscript meet PLOS Global Public Health’s publication criteria ? Is the manuscript technically sound, and do the data support the conclusions? The manuscript must describe methodologically and ethically rigorous research with conclusions that are appropriately drawn based on the data presented.

Reviewer #2: Yes

Reviewer #3: Yes

Has the statistical analysis been performed appropriately and rigorously?

Reviewer #2: Yes

Reviewer #3: Yes

Have the authors made all data underlying the findings in their manuscript fully available (please refer to the Data Availability Statement at the start of the manuscript PDF file)?

Reviewer #2: No

Reviewer #3: Yes

Is the manuscript presented in an intelligible fashion and written in standard English?

Reviewer #2: Yes

Reviewer #3: Yes

Review Comments to the Author

Reviewer #2: Thank you for the opportunity to review the manuscript “Distribution and transmission of M. tuberculosis in a high-HIV prevalence city in Malawi: a genomic and spatial analysis”. I have reviewed a revised version of this manuscript and, as such, have had access to the feedback from previous reviewers and the responses from the authors.

This study leverages high-resolution surveillance data and whole genome sequences; and employs sophisticated spatial and transmission modelling methods to describe the patterns of TB transmission in Blantyre (a city with high burden of TB and HIV-associated TB).

The authors demonstrate that TB transmission in Blantyre is predominantly widespread throughout the city, but this pattern overlaps with multi-focal pockets of hyper-local transmission. These findings potentially translate into policy with the authors proposing that TB control efforts should primarily be applied city wide, rather than spatially-targeted.

I applaud the authors for their considerable efforts. However, I feel there are some major weaknesses with the study and some opportunities for potential revision.

General weaknesses:

These have been identified by previous reviewers and are partially addressed by the authors.

1) Attrition.

Only 6% of the notified cased during the study period were ultimately included in the transmission analysis. Although many of the cases not included represent culture-negative TB, only 18% of culture-positive case could be included in the analysis. This hugely threatens the validity of the transmission inferences the authors make (as these methods assume dense sampling of outbreaks) and the extension of these into TB control recommendations. For instance, it may be that many of the unclustered isolates (46% of sequenced isolates) were only unclustered because the cases within their transmission networks were upsampled. In the extreme (although this is an unlikely scenario), it may be that there are in fact multiple spatially-conserved transmission clusters that were undetected as they were upsampled. Were these hypothetical clusters detected, the conclusion may have been the opposite, and the authors may have recommended spatially-targeted active case findings for primary TB control efforts. Ultimately, while the findings of the study align with our working hypothesis about the nature of TB transmission in high burden settings (that it is widespread), the data it draws its conclusions from are too weak for the study to be confirmatory and it remains hypothesis generating; and its impact is limited.

2) Home addresses

The analysis is limited by the reliance on home addresses only for the mapping of spatial foci of transmission, incorporation of spatial correlation into regression models, the use of spatial covariates in the models assessing predictors being in a transmission clusters/genetic relatedness and the analysis of transmission flow across districts. Indeed, the authors already allude to the small role household transmission likely plays in this setting, emphasising the importance of transmission in areas other than home addresses. It is unfortunate that no data on mobility patterns, workplace addresses etc were not available.

Points for revision:

1) Was any ascertainment bias correction performed for the phylogenies, particularly for the BEAST components, given the inputs were multi-sequence alignments of SNPs rather than whole genomes? If not, the branch lengths and dates for the branching of timed trees may be incorrect. Providing the BEAST input XML file may demonstrate this in addition to improving the reproducibility of the analysis.

2) The authors demonstrate the sharing a TB clinic is associated with a higher odds of a pair of cases being in the same transmission network; and (for network #2) that this is associated with a higher risk that a pair of cases are closer genetically related. I am unsure of what to make of this. Perhaps the authors can expand the rationale for including this is a covariate and providing some interpretation of its significance. For example, is this intended as a proxy for people sharing the same geographic region (if so, is this colinear with distance between home addresses)? Or do the authors think it is causally related to being in a transmission network in some other way? Does this finding have implications for TB control?

3) For the regression analysis investigating the outcome of pairs of cases belonging to the same transmission network: What is the rationale for only including clustered cases (n=389) rather than all cases (707)? Does this potentially bias these results? For example what if many of the unclustered cases share the same TB clinics as those in putative transmission networks and their exclusion inflated the effect size of the association between sharing a TB clinic and belonging to the same transmission network?

4) In Figure 5B, there are no lines for the estimates for same TB clinic and same HIV clinic for network #43

5) The data availability statement does not currently specify any publicly available deposition. While it does not appear to be mandated by PLOS GPH , the publishing of the analysis code would greatly enhance the reproducibility of the work and offers an opportunity for others to learn from, what is in my opinion the most impactful part of this work, the methods.

Conclusion:

I commend the authors for their substantial efforts in collecting and harmonising multi-modal data sources and for the rigorous application of the methods employed in this study. However, practical challenges related to data availability have significantly constrained the novelty and overall impact of the work. Despite these limitations, the study remains a commendable contribution to the field.

I would like to raise an important issue regarding equity in global health research, particularly given PLOS Global Public Health's mission statement: “PLOS Global Public Health addresses deeply entrenched global inequities in public health...” It was disappointing to note that only 3 out of 18 contributing institutions were Malawian, and just 6 of the 24 authors were affiliated with Malawian institutions, with the majority of representation coming from the Global North. I recognise the complexities of funding, study design, data analysis, and authorship dynamics are unique and cannot be discerned from an author list alone. However, equity should remain a fundamental consideration for global health researchers. Moving beyond a culture of "parachute research" is essential for ensuring meaningful collaboration and representation. I do not intend to be accusatory but rather to emphasise the ongoing relevance of global heath equity going forward. This issue has been extensively discussed both in this journal (e.g., DOI:10.1371/journal.pgph.0000160, DOI:10.1371/journal.pgph.0003141) and elsewhere (DOI:10.1016/S0140-6736(24)02323-7).

Reviewer #3: Chitwood and colleagues have conducted an incredibly sophisticated analysis of a complex problem. Using routinely collected electronic health data and whole genome sequencing, they have conducted an insightful analysis of TB transmission in Blantyre, with findings that are relevant to policymakers. I note the prior comments from four peer-reviewers via another journal have already been answered adequately. The manuscript is well-written and conclusions are not overstated. I have the following additional suggestions/requests for clarification which are largely minor:

1. Suggest make it clear in methods that the only locations being considered for the spatial analysis were the household, the TB clinic or the HIV clinic. This does also come through clearly in the discussion section.

2a. Suggest make it clear in the figures (e.g. figure 1A/B) which location is being referred to e.g. that the location being highlighted is the household (if I understood this correctly) rather than the location of their clinic?

2b. For Figure 1B, is this HIV prevalence only with regards to the TB cases in dataset, or Blantyre-wide estimates independent of the study? Suggest make clear in the figure's title.

3. Within the results section "Whole genome sequencing analysis", suggest include denominators with the various numerators so that the percentages are clear to the reader. For example, to reflect on the fact that only estimated 8% of all those who could have potentially been included in study dataset landed up having matched sequencing and clinical/epi data.

4a. I appreciate the addition of table 1 based on prior peer-review. Does the available data allow for an additional column of all Blantyre TB cases, to compare between the overall culture-positive TB cases and the ones included in this study?

4b. Furthermore, in line with the main conclusions of the study, could additional rows be added with regards to the presence of a household contact, and with regards to zone inhabited, to know whether either of these were under- or over-represented in their study sample?

5a. My understanding is that clinic sputum culture samples were supplemented by research tests. In response to a reviewer's comments, it was noted that sequencing was performed on the research tests. Suggest include context of how samples were collected or how they fit within the algorithm of routine testing in Malawi.

5b. With this, could the authors provide context on whether all included samples were sourced from passive case finding and individuals self-presenting to their TB clinics? And thereby a pool of infectious (possibly asymptomatic) TB disease is thereby not considered in this dataset?

6. Finally, I also think there is a small typo in this sentence from methods section "These data were combined in a Bayesian model to derive*d* highly spatially resolved HIV prevalence estimates, as described previously"

7. PLOS authors have the option to publish the peer review history of their article (what does this mean? ). If published, this will include your full peer review and any attached files.

**Do you want your identity to be public for this peer review?** For information about this choice, including consent withdrawal, please see our Privacy Policy .

Reviewer #2: No

Reviewer #3: No

---

## [Editor Report · Decision Letter 2]

21 Jan 2025

Distribution and transmission of M. tuberculosis in a high-HIV prevalence city in Malawi: a genomic and spatial analysis

PGPH-D-24-01419R2

Dear Ms. Chitwood,

We are pleased to inform you that your manuscript 'Distribution and transmission of M. tuberculosis in a high-HIV prevalence city in Malawi: a genomic and spatial analysis' has been provisionally accepted for publication in PLOS Global Public Health.

Best regards,

Madhukar Pai, MD, PhD

Editor-In-Chief
